# Non-invasive vocal biomarker is associated with pulmonary hypertension

Jaskanwal Deep Singh Sara[1], Elad Maor[2,3], Barry Borlaug[1], Bradley R. Lewis[4], Diana Orbelo[5], Lliach O. Lerman[6], Amir Lerman[1]*

1 Department of Cardiovascular Diseases, Mayo College of Medicine, Rochester, MN, United States of America, 2 Chaim Sheba Medical Center, Tel Hashomer, Israel, 3 Sackler School of Medicine, Tel Aviv University, Tel Aviv, Israel, 4 Division of Biomedical Statistics and Informatics, Mayo College of Medicine, Rochester, MN, United States of America, 5 Divison of Laryngology, Mayo College of Medicine, Rochester, MN, United States of America, 6 Division of Nephrology and Hypertension, Mayo Clinic, Rochester, MN, United States of America

* lerman.amir@mayo.edu

**Data Availability Statement:** All relevant data are within the paper and its Supporting Information files.

**Funding:** This study was in part funded by Beyond Verbal Communications. The funders had no role

## Abstract

Emerging data suggest that noninvasive voice biomarker analysis is associated with coronary artery disease. We recently showed that a vocal biomarker was associated with hospitalization and heart failure in patients with heart failure. We evaluate the association between a vocal biomarker and invasively measured indices of pulmonary hypertension (PH). Patients were referred for an invasive cardiac hemodynamic study between January 2017 and December 2018, and had their voices recorded on three separate occasions to their smartphone prior to each study. A pre-established vocal biomarker was determined based on each individual recording. The intra-class correlation co-efficient between the separate voice recording biomarker values for each individual participant was 0.829 (95% CI 0.740–0.889) implying very good agreement between values. Thus, the mean biomarker was calculated for each patient. Patients were divided into two groups: high pulmonary arterial pressure (PAP) defined as $\geq$ 35 mmHg (moderate or greater PH), versus lower PAP. Eighty three patients, mean age 61.6 ± 15.1 years, 37 (44.6%) male, were included. Patients with a high mean PAP ($\geq$ 35 mmHg) had on average significantly higher values of the mean voice biomarker compared to those with a lower mean PAP (0.74 ± 0.85 vs. 0.40 ± 0.88 p = 0.046). Multivariate logistic regression showed that an increase in the mean voice biomarker by 1 unit was associated with a high PAP, odds ratio 2.31, 95% CI 1.05–5.07, p = 0.038. This study shows a relationship between a noninvasive vocal biomarker and an invasively derived hemodynamic index related to PH obtained during clinically indicated cardiac catheterization. These results may have important practical clinical implications for telemedicine and remote monitoring of patients with heart failure and PH.

## Introduction

Heart failure (HF) affects over 6 million people in the United States and results in more than 1 million hospitalizations per year [1]. In patients aged 65 years and older, there are more

in study design, data collection and analysis, decision to publish, or preparation of the manuscript. Dr. Elad Maor serves as a consultant for Beyond Verbal Communications. This author received no specific funding for this work. The remaining authors have nothing to disclose

**Competing interests:** Dr. Elad Maor serves as a consultant for Beyond Verbal Communications. This author receives no salary for this work. This does not alter our adherence to PLOS ONE policies on sharing data and materials. The remaining authors have nothing to disclose.

**Abbreviations:** CI, Cardiac index; HF, Heart failure; PAP, Pulmonary artery pressure; PCWP, Pulmonary capillary wedge pressure; PH, Pulmonary hypertension; PVR, Pulmonary vascular resistance; RA, Right atrial.

hospitalizations for a primary diagnosis of HF than any other condition [2]. HF is a debilitating disease and is associated with significant morbidity and mortality, re-hospitalization, and costs to health care systems [3]. Despite advances in patient care, the incidence of adverse outcomes following hospitalizations remains high among patients with HF [4, 5].

Pulmonary hypertension (PH) is highly prevalent in patients with HF regardless of ejection fraction [6, 7]. Irrespective of the cause, PH is a marker of disease severity and is related to more severe symptoms, worse exercise tolerance, higher hospitalization rates and a greater likelihood to require cardiac transplantation [8]. In addition, PH associated with left HF is associated with higher mortality in both reduced and preserved ejection fraction states [9–12]. Hemodynamic indices related to PH measured invasively at cardiac catheterization assist in determining HF severity, predict outcomes such as HF-related hospitalization and death [13–16], and could therefore help influence decisions in modifying therapy.

Monitoring the impact of therapy remotely holds the potential to reduce HF-related hospitalizations, improve quality of life and optimize the use of the limited resources [17]. Data from clinical trials investigating telemedicine-based interventions are promising, showing improvements in quality of life, and reductions in HF-related hospitalizations and all-cause mortality [18–22]. For example, wireless implantable hemodynamic monitoring systems of pulmonary arty pressure (PAP) allowed better remote HF management and reduced hospitalization rates [23, 24]. Voice signal is an emerging non-invasive biomarker that has been associated with a number of disease states [25, 26]. We previously identified a significant relationship between specific vocal biomarkers and coronary artery disease, underscoring the potential utility of voice signal analysis in identifying individuals with cardiovascular disease [27]. We recently extended these observations by showing that a pre-specified voice biomarker was associated with increased mortality and re-hospitalization in patients with HF [28] (manuscript in press). Thus in the current study we hypothesized that the same voice biomarker might also be related to hemodynamic indices reflective of pulmonary vascular disease, obtained invasively in patients referred for clinically indicated cardiac catheterization hemodynamic studies.

## Materials and methods

### Study population

The study population consisted of consecutive patients who were referred for an elective clinically indicated invasive cardiac catheterization hemodynamic study. Patients were enrolled between January 1, 2017, and December 31, 2018. Those with a history of heart transplant were excluded from the study, as were those who were pregnant, aged less than 18 years, and individuals who had a current or known history of a primary voice disorder. The study protocol was approved by the institutional review board at Mayo Clinic, and all patients provided informed consent for participation.

### Voice characteristics

Following enrollment and before the planned cardiac catheterization, each study participant was asked to speak aloud into a recording device. Recording was performed by the patients with no prior coaching or training. The voice was recorded, stored online, and analyzed for multiple features of voice intensity and frequency using "*Vocalis's*" clinical trial application, which was downloaded to the patients' personal smartphone [25]. Voice analysis was blinded with respect to patient identifiers and clinical information, and was done in a semi-automated fashion. To maintain high quality recordings, all voice files were examined by a voice analytics expert, after which a defined set of acoustic features were extracted from each voice file. Thus,

no editing or subjective interpretation was required in the process. Voice recordings of poor quality, typically from excessive background noise or multiple voices being recorded, precluded voice feature analysis and were excluded. Study participants underwent a total of three 30-second separate baseline voice recordings for analysis: R1—participants were asked to read a pre-specified text; R2—participants were asked to describe a positive emotional experience; and R3—participants were asked to describe a negative emotional experience, as previously described [27].

The vocal biomarker used in the current analysis was developed by "Vocalis" using voice processing techniques. The biomarker was developed with the help of a cohort of chronic patients who were registered to a call center in Israel (N = 10,583). In brief, a total of 223 acoustic features were extracted from 20 seconds of speech for each patient. The Mel Frequency Cepstral Coefficients were used to extract information from the voice signal [29], and represent a sound processing tool that is used for voice recognition and for automatic classification between healthy and impaired voices [30–32]. The input for computation of the Mel Frequency Cepstral Coefficients is a speech signal that is further analyzed using the Fourier transform mathematical function. Acoustic features extracted included the following, as previously described: Mel Cepstrum representation, Pitch and Formant Measures, Jitter, Shimmer and Loudness [17]. The voice biomarker is a unitless unbounded scalar, which is a linear combination of the 223 acoustic features mentioned above. The biomarker was calculated based on this cohort with the use of machine learning and artificial intelligence techniques, and its prediction capabilities were estimated based on the biomarker's hazards ratio and $p$ value, with respect to overall survival. Preliminary data suggested that this biomarker is associated with adverse outcome among patients with congestive HF [28].

## Study endpoint

The primary end points of the current study was a diagnosis of moderate or greater PH (defined as a mean PAP $\geq$ 35 mmHg) obtained from the invasive cardiac hemodynamic study, as well as other measurements reflecting the severity of pulmonary vascular disease including pulmonary vascular resistance (PVR, Wood Units), and pulmonary capillary wedge pressure (PCWP, mmHg). As additional end-points, we also included other measurements obtained at the index invasive cardiac catheterization hemodynamic study including right atrial (RA) pressure measured in mmHg, and cardiac index (CI) measured in L/minute. All invasive hemodynamic measurements and data obtained at each cardiac catheterization were determined by the operating physician who was blinded to patient voice data.

## Statistical analysis

Data are presented as a mean ± standard deviation for normally distributed continuous variables, and as frequency (%) for categorical variables. In the primary analysis, the study population was divided a priori into two groups: those with a high mean PAP (defined as $\geq$ 35 mmHg) versus those with a lower mean PAP ($<$ 35 mmHg). This threshold was selected as a mean PAP $\geq$ 35 mmHg has been traditionally categorized as moderate or greater PH [33]. As part of secondary analyses, individuals were separately grouped into those with an high versus lower PVR, with an high PVR defined as $\geq$ 1.7 Wood Units, which is 2 standard deviations greater than normal [9]; high versus lower PCWP, with a high PCWP defined with the conventional threshold of $\geq$ 15 mmHg to distinguish post- versus pre-capillary PH; and high versus lower RA pressure and high versus lower CI by dividing individuals into the highest tertile versus the lower two tertiles according to the statistical distribution for each measurement.

After excluding poor quality recordings, values of the pre-specified voice biomarker were obtained from each high quality recording for each patient. In our recent study in which we showed that the same pre-specified voice biomarker was associated with increased mortality and re-hospitalization in patients with HF [28] (manuscript in press), we did not show any significant differences in the association between the voice biomarker and clinical outcomes when the voice biomarker used was derived from individuals recording their voices talking about positive, negative, or neutral experiences separately. Consequently, we elected to determine the agreement across the separate voice biomarker values for each individual participant by calculating the intra-class correlation coefficient with 95% confidence interval. We then calculated the mean voice biomarker value for each patient and used these numbers in our final analyses. In cases in which a study participant had one or more voice recording samples excluded due to poor quality, the remaining high-quality samples were retained and used to determine the mean biomarker value for that patient. Normal distribution and equal variance were checked by the Shapro-Wills test, and Levene's test respectively for each variable. The mean biomarker values were then compared between groups using Student's t-test. The same dataset was used for all analyses. Univariate logistic regression analyses were undertaken to determine the association between the mean voice biomarker, as the independent variable, and each of the following individually as categorical dependent variables: a high PAP, PVR, PWCP, RA pressure, and CI. Each association was examined in all patients and after stratifying by a high versus lower PCWP. The distinction between high and lower PCWP was chosen to distinguish PH that was "post-capillary" in etiology and therefore related to coexisting left HF versus that which was "pre-capillary" in etiology and therefore related to a primary vascular and/or lung pathology. Finally, multivariable logistic regression analyses were undertaken to determine the relationships between the mean voice biomarker value, as the independent variable, and each of the following individually as categorical dependent variables: a high PAP, PVR, PWCP, RA pressure, and CI. Each association was examined in all patients and after stratifying by a high versus lower PCWP. Each analysis was adjusted for age, sex, hypertension, diabetes mellitus, and NYHA class as these factors are known to be associated with PH and/or HF and could therefore act as potential confounders. For all the above analyses, the type 1 error rate was 0.05 in a 2-sided test and p values and confidence intervals were calculated and presented at the 95% confidence level. The statistical analyses were performed using JMP 9 software (SAS Institute, Inc., Cary, NC, USA).

## Results

### Study population

The study population included a total of 99 patients who were enrolled between January 1, 2017, and December 31, 2018, all of whom underwent a clinically indicated invasive cardiac catheterization hemodynamic study. Of these, 16 (16.2%) individuals had a history of heart transplant and were excluded from the final analysis. Thus the final study sample included 83 patients (mean ± standard deviation age of 61.6 ± 15.1 years) 37 (44.6%) of whom were male. Twenty one (25.3%) patients had a diagnosis of diabetes mellitus, 53 (63.9%) had hypertension, 44 (53.0%) had dyslipidemia, 46 (55.4%) had a body mass index of greater than 30 kg/m$^2$, 2 (2.4%) were current smokers and 36 (43.4%) were former smokers. Two (2.4%) patients had NYHA class I symptoms, 20 (24.1%) had class II, 42 (50.6%) had class III, and 19 (22.9%) had class IV symptoms. Forty three (51.8%) had an estimated glomerular filtration rate of less than 60 mL/minute, and 14 (16.9%) had an ejection fraction of less than 40% on echocardiogram. Common diagnoses following the invasive hemodynamic study were PH, systolic HF and a normal study in 35 (42.2%), 18 (21.7%) and 8 (9.6%) patients, respectively.

## Baseline characteristics

Each of the 83 study subjects undertook three separate voice recordings giving rise to 249 potentially analyzable voice samples, of which 243 (97.6%) were adequate for voice feature extraction and analysis. The remaining recordings were excluded from the final analysis as background noise or multiple voices precluded analysis and voice feature extraction. Patients were a priori divided into those with versus those without moderate or greater PH defined as a mean PAP $\geq$ 35 mmHg at the index invasive hemodynamic study. Table 1 compares the baseline characteristics between the two groups. There were no significant differences in demographic or clinical variables between groups, nor was there a significant difference in the frequency of medication use.

## Relationship between voice biomarker and pulmonary hypertension

In addition to dividing patients according to the mean PAP, subjects were also dichotomized as having high versus lower values for the other hemodynamic measurements evaluated during the index hemodynamic study as follows: a priori high ($\geq$ 1.7 Wood Units) versus lower PVR; a priori high ($\geq$ 15 mmHg) versus lower PCWP, with a PCWP of $\geq$ 15 mmHg; and post-hoc

**Table 1. Baseline characteristics of the study cohort.**

|  | Pulmonary Arterial Pressure $\geq$ 35mmHg, N = 27 (32.5%) | Pulmonary Arterial Pressure $<$ 35mmHg, N = 56 (67.5%) | P value |
|---|---|---|---|
| Age $\pm$ SD (years) | 65.4 $\pm$ 17.4 | 59.8 $\pm$ 13.7 | 0.154 |
| Male (%) | 11 (40.7) | 26 (46.4) | 0.625 |
| Hypertension (%) | 19 (70.4) | 34 (60.7) | 0.387 |
| Diabetes Mellitus (%) | 10 (37.0) | 11 (19.6) | 0.094 |
| Hyperlipidemia (%) | 17 (63.0) | 27 (48.2) | 0.205 |
| BMI $\pm$ SD (kg/m$^2$) | 33.9 $\pm$ 7.0 | 31.9 $\pm$ 9.6 | 0.291 |
| Smoking Status |  |  |  |
| • Never (%) | 14 (51.9) | 31 (55.4) | 0.100 |
| • Former (%) | 11 (40.7) | 25 (44.6) |  |
| • Current (%) | 2 (7.4) | 0 (0.0) |  |
| Ejection Fraction $\pm$ SD (%) | 56.6 $\pm$ 15.2 | 52.0 $\pm$ 16.5 | 0.218 |
| NYHA Class |  |  |  |
| • Class I (%) | • 1 (3.7) | • 1 (1.8) | 0.913 |
| • Class II (%) | • 6 (22.2) | • 14 (25.0) |  |
| • Class III (%) | • 13 (48.2) | • 29 (51.8) |  |
| • Class IV (%) | • 7 (25.9) | • 12 (21.4) |  |
| eGFR $\pm$ SD (mL/minute per 1.73 m$^2$) | 58.3 $\pm$ 24.7 | 59.8 $\pm$ 18.3 | 0.780 |
| ACE-Inhibitors/Angiotensin Receptor Blockers (%) | 8 (29.6) | 27 (48.2) | 0.104 |
| Beta-blocker (%) | 16 (59.3) | 28 (50.0) | 0.427 |
| Aldosterone Antagonists (%) | 4 (14.8) | 15 (26.8) | 0.211 |
| Dihydropyridines (%) | 5 (18.5) | 12 (21.4) | 0.757 |
| Endothelin Receptor Antagonists (%) | 0 (0.0) | 2 (3.6) | 0.206 |
| Aspirin (%) | 8 (29.6) | 29 (51.8) | 0.054 |
| Phosphodiesterase Inhibitors (%) | 1 (3.7) | 4 (7.1) | 0.521 |
| Riociguat (%) | 0 (0.0) | 1 (1.8) | 0.373 |
| Prostacyclin (%) | 1 (3.7) | 1 (1.8) | 0.605 |

Abbreviations: BMI–body mass index; eGFR–estimated glomerular filtration rate; NYHA–New York Heart Association.

high ($\geq$ 10 mmHg) versus lower RA pressure and high ($\geq$ 3 L/minute) versus lower CI after dividing individuals into the highest tertile versus the lower two tertiles according to the statistical distribution for each measurement.

A pre-established vocal biomarker was determined based on each separate recording for each individual participant. The intra-class correlation co-efficient between the separate voice recording biomarker values for each individual participant was 0.829 (95% CI 0.740–0.889) implying very good agreement between values. Thus, the mean biomarker was calculated for each patient. Patients with a high mean PAP had significantly higher mean values of the voice biomarker compared to those with a low mean PAP (0.74 ± 0.85 vs. 0.40 ± 0.88 p = 0.046) (Fig 1A). After stratifying patients by a high and low PCWP, there was no significant difference in values of the mean voice biomarker between patients with a high versus low mean PAP in either patients with a high or low PCWP (Fig 1B and 1C). Values of the mean voice biomarker did not vary significantly between patients with a high versus low PVR amongst all patients (Fig 2A) or in patients stratified by high and low PCWP (Fig 2B and 2C). Similarly the value of the mean voice biomarker did not vary significantly between patients with a high versus low PCWP, high versus low RAP, and high versus low CI amongst all patients.

### Univariable and multivariable analyses

Table 2 shows the results of a univariable analysis evaluating the association between hemodynamic indices and the mean voice biomarker among all patients and after stratifying by high

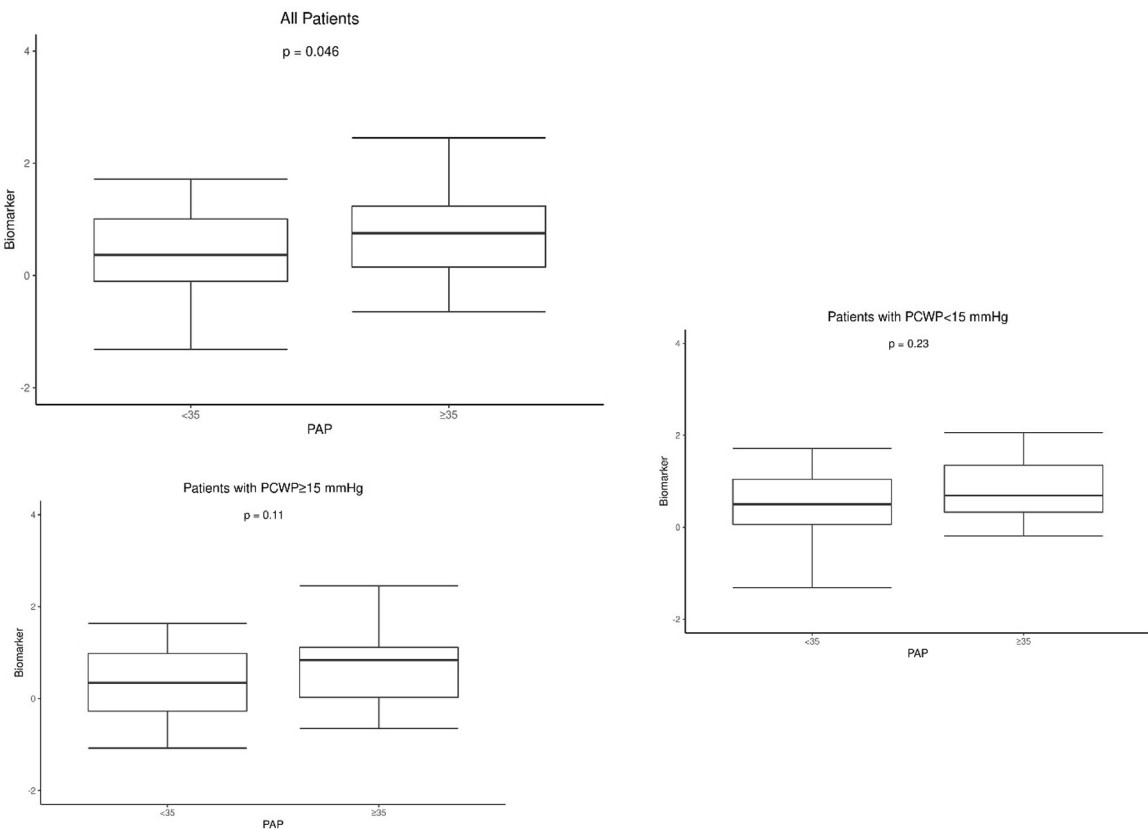

**Fig 1. Boxplots comparing values of the voice biomarker between individuals with a high ($\geq$ 35 mmHg) versus lower pulmonary arterial pressure.** A: In all patients; B: In patients with a PCWP $\geq$ 15 mmHg; C: In patients with a PCWP < 15 mmHg. Abbreviations: PAP–Pulmonary arterial pressure; PWPW–Pulmonary capillary wedge pressure. *statistically significant difference between groups.

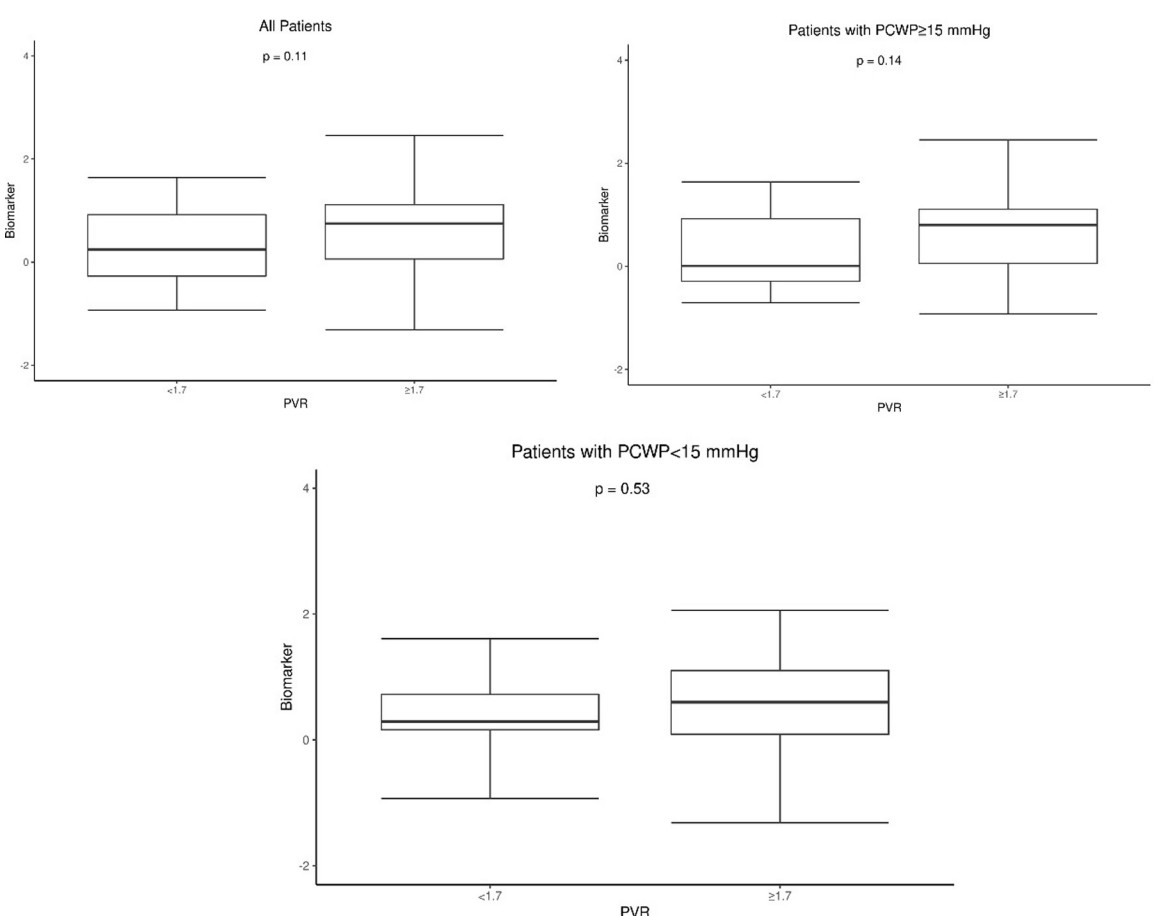

**Fig 2. Boxplots comparing values of the voice biomarker between individuals with a high ($\geq$ 1.7 Wood Units) versus lower pulmonary vascular resistance.** A: In all patients; B: In patients with a PCWP $\geq$ 15 mmHg; C: In patients with a PCWP < 15 mmHg. Abbreviations: PVR–Pulmonary vascular resistance; PWPW–Pulmonary capillary wedge pressure. *statistically significant difference between groups.

versus low PCWP. In all patients, the mean voice biomarker was significantly associated with a mean PAP $\geq$ 35 mmHg such that an increase in the mean biomarker by 1 unit was associated with an odds ratio (95% CI) of 1.92 (1.00–3.65, p = 0.049). This association did not persist after stratifying by high or low PCWP. In a separate univariable analysis, the biomarker derived

**Table 2. Univariate analyses evaluating the association between mean voice biomarker and hemodynamic indices measured at invasive hemodynamic study.**

| | | Odds Ratio for Association with Voice Biomarker | 95% Confidence Interval | P value |
|---|---|---|---|---|
| **Mean Pulmonary Arterial Pressure $\geq$ 35 mmHg** | *All* | 1.92 | 1.00–3.65 | 0.049* |
| | *PCWP $\geq$ 15mmHg* | 1.89 | 0.87–4.12 | 0.109 |
| | *PCWP < 15mmHg* | 2.09 | 0.64–6.82 | 0.223 |
| **Pulmonary Vascular Resistance $\geq$ 1.7 Wood Units** | *All* | 1.79 | 0.88–3.65 | 0.110 |
| | *PCWP $\geq$ 15mmHg* | 2.06 | 0.81–5.24 | 0.130 |
| | *PCWP < 15mmHg* | 1.45 | 0.48–4.43 | 0.513 |

Abbreviations–PCWP: pulmonary capillary wedge pressure;

*statistically significant difference between groups.

while describing a negative experience was significantly associated with a high mean PAP, odds ratio (95% CI) 1.91 (1.05–3.49, p = 0.035) but not when the biomarker was derived while describing a neutral experience, odds ratio (95% CI) 1.31 (0.69–2.49, p = 414) or positive experience, odds ratio (95% CI) 1.53 (0.85–2.76, p = 016). The mean voice biomarker was not significantly associated with a PVR ≥ 1.7 Wood Units amongst all patients or in those after stratifying by high and low PCWP. Further, the voice biomarker was not significantly associated with a high PVR when it was derived in individuals describing a negative experience, odds ratio (95% CI) 1.54 (0.88–2.71, p = 0.13), neutral experience, odds ratio (95% CI) 1.57 (0.76–3.23, p = 0.223) or positive experience, odds ratio (95% CI) 1.44 (0.75–2.75, p = 0.272). The voice biomarker was not significantly associated with any other hemodynamic index.

Table 3 shows the results of multivariable analyses evaluating the association between hemodynamic indices and the mean voice biomarker among all patients and after stratifying by high versus low PCWP, after adjusting for age, sex, hypertension, diabetes mellitus, and NYHA class. In all patients, the voice biomarker was significantly associated with a mean PAP ≥ 35 mmHg such that an increase in the biomarker by 1 unit was associated with an odds ratio (95% CI) of 2.31 (1.05–5.07, p = 0.038). This association persisted with borderline significance amongst individuals with a high PCWP, odds ratio (95% CI) 2.72 (0.96–7.68, p = 0.06), but not in those with a low PCWP. In addition, the mean voice biomarker was associated with a PVR ≥ 1.7 Wood Units with borderline significance such that an increase in the biomarker by 1 unit was associated with an odds ratio (95% CI) of 2.14 (0.94–4.87, p = 0.07). This association was statistically significant amongst individuals with a high PCWP, odds ratio (95% CI) 3.86 (1.07–13.91, p = 0.039), but not in those with a low PCWP.

There were no significant associations between the voice biomarker and other hemodynamic indices among all patients and after stratifying by PCWP in univariate or multivariate analyses.

## Discussion

### Summary of findings

In the current study we demonstrate, for the first time, an association between a non-invasive voice biomarker and hemodynamic indices measured invasively at cardiac catheterization in an unselected sample of patients referred for a clinically indicated invasive hemodynamic study. Specifically we show that the mean voice biomarker was associated with measurements related to PH and pulmonary vascular disease, and had significantly higher values in individuals with a high mean PAP. Further, in univariable and multivariable analyses adjusting for age, sex, hypertension, diabetes mellitus, and NYHA classification the mean voice biomarker was

**Table 3. Multivariable analyses evaluating the association between mean voice biomarker and hemodynamic indices measured at invasive hemodynamic study.**

|  |  | Odds Ratio for Association with Voice Biomarker | 95% Confidence Interval | P value |
|---|---|---|---|---|
| **Mean Pulmonary Arterial Pressure ≥ 35 mmHg** | *All* | 2.31 | 1.05–5.07 | 0.038* |
|  | *PCWP ≥ 15mmHg* | 2.72 | 0.96–7.68 | 0.060 |
|  | *PCWP < 15mmHg* | 2.42 | 0.62–9.50 | 0.206 |
| **Pulmonary Vascular Resistance ≥ 1.7 Wood Units** | *All* | 2.14 | 0.94–4.87 | 0.070 |
|  | *PCWP ≥ 15mmHg* | 3.86 | 1.07–13.91 | 0.039* |
|  | *PCWP < 15mmHg* | 1.66 | 0.39–7.03 | 0.493 |

Abbreviations–PCWP: pulmonary capillary wedge pressure;

*statistically significant difference between groups; ‡ Multivariate analyses adjusted for age, sex, hypertension, diabetes mellitus, and New York Heart Association class.

significantly associated with a high mean PAP and, with borderline significance, a high PVR in all patients. Finally, after stratifying by high versus low PCWP to distinguish between individuals with "post-capillary" PH related to left sided-HF versus isolated "pre-capillary" PH unrelated to left sided-HF respectively, we showed that the mean voice biomarker was associated with a high mean PAP and with a high PVR in individuals with "post-capillary" PH after adjusting for co-variables. Thus, by demonstrating a relationship between invasively obtained hemodynamic measurements related to PH known to be associated with adverse clinical outcomes [15, 16], and a non-invasive voice biomarker, the current study supports the potential role for identifying at-risk patients with HF using voice signal analysis, or potentially, using voice analysis to detect hemodynamic changes in patients with established HF or PH.

## Potential mechanism underling the relationship between voice signal analysis and heart failure

Voice signal analysis is an emerging non-invasive biomarker that has been associated with a number of disease states including autistic spectrum disorders, Parkinson's disease, and other neurologic disorders [25, 26]. We previously studied subjects who underwent clinically indicated coronary angiography who had their voices recorded to their personal smartphone devices using the "*Vocalis*" application, and identified two voice features that were associated with the presence of coronary artery disease [27]. We recently extended these observations to an additional facet of cardiovascular disease in a recent study by showing that the same pre-specified vocal biomarker used in the current study was associated with increased mortality and re-hospitalization in patients with HF [28] (manuscript in press). Similarly, in another study, the voices of ten patients with decompensated HF were analyzed during acute treatment and the authors showed a correlation between several voice markers and improvement in HF symptoms [34]. In the current study we build further upon these findings by showing that the same pre-specified voice biomarker investigated in our previous study [28] was associated with hemodynamic indices relevant to PH that are known to predict outcomes in HF, namely mean PAP and PVR [15, 16], in a stable group of patients who underwent clinically indicated invasive hemodynamic studies.

Epidemiologically, HF arises most commonly as a consequence of ischemic heart disease [35], which itself is typically related to atherosclerosis, widely considered a systemic inflammatory disorder. Consequently, coronary artery disease is often associated with atherosclerotic disease in other vascular beds leading to cerebrovascular disease, vascular dementia, retinopathy, peripheral arterial disease, and chronic kidney disease. Thus, the findings that voice signal characteristics are associated with coronary artery disease as in our previous study [27], and with invasively measured indices relevant to PH as in the current study, could relate to the systemic nature of atherosclerosis and/or inflammation more generally and their established effects on the vasculature of the heart, and potentially the less well established effects on the vasculature which perfuse organs of phonation. Additionally, the vagus nerve participates in voice production together with other cranial nerves, whilst also playing a critical role in autonomic regulation of the heart through its superior, inferior, and thoracic branches. The vagus nerve is also associated with heart rate control and variability, which has a well-established relationship with coronary artery disease [36] and cardiovascular events [37]. Thus the unifying relationship between voice signal characteristics and cardiac health could be neurally mediated either directly, or indirectly through the effects of coronary disease. Alternatively, PH-related pulmonary arterial dilatation might lead to partial compression of the left recurrent laryngeal nerve as it circles around the aorta and between the great vessels, akin to a variant of Ortner's syndrome in which patients present with voice hoarseness. The current study did not

investigate a biologic mechanism for the observed associations, and further studies are required to better understand the precise relationship between cardiac function and voice characteristics.

Emotional disturbance and stress more generally are known risk factors for coronary artery disease and cardiovascular disease [38], which may in part be explained by the relationship between mental stress and the adrenergic system [39]. Further, HF is characterized by a well-described and predictable constellation of pathophysiological changes including impaired myocardial contractility and/or relaxation, diminished cardiac output, increasing filling pressures and myocardial remodeling along with circulatory changes influenced by upregulation of the renin-angiotensin-aldosterone system and sympathetic nervous system in an attempt to preserve end organ-perfusion. Thus alterations in the functioning of the adrenergic nervous system are expected in patients with HF. Additionally, emotional stress has been shown to change human voice characteristics, including an increase in fundamental frequency [40, 41]. Thus, clinically measurable changes in cardiac structure and function that are underpinned in part by alterations in adrenergic nervous system functioning may occur in parallel with vocal changes captured using voice signal analysis, which are themselves influenced by the complex interplay between the sympathetic nervous system and emotional stress. It is, therefore, possible that our voice analysis system indirectly assesses the state of the sympathetic nervous system and hereby provides a means to quantify stress and identify patients at risk of cardiac disease, and more specifically, high risk of adverse outcomes in individuals with HF. Interestingly, we did show that when the voice biomarker was derived from individuals describing a negative experience only there was an association between the voice biomarker and a high mean PAP, suggesting that emotional stress may indeed have a role in the complex interplay between the sympathetic nervous system, cardiovascular disease, and phonation. These hypotheses require greater clarification with further studies.

## Clinical implications in patients with heart failure

PH is prevalent in patients with reduced and preserved ejection fraction HF [6, 7]. So-called "post-capillary" PH represents the most common form of PH [9], and is a marker of disease severity relating to more severe symptoms, worse tolerance to exercise, higher hospitalization rates and a greater likelihood to require cardiac transplantation [8], as well as higher mortality rates [9–12]. Hemodynamic indices measured invasively at cardiac catheterization such as mean PAP can quantify the severity of PH and HF, represent the gold-standard in the assessment of these parameters, and can predict outcomes such as HF-related hospitalization and death [13–16]. Such assessments however are limited by their invasive nature and the need for patient visits. The current study demonstrates an association between a non-invasively measured voice biomarker that can be obtained remotely, and invasive measurements of PH (PAP and PVR) that have a known role in predicting adverse outcomes in HF patients [15, 16]. Thus the current study supports the potential role of remote monitoring of HF patients using voice signal analysis to identify those who could have high mean PAP and/or PVR portending higher risk. Such a strategy could be used to stratify patients according to risk and recommend more frequent in-person assessments as appropriate.

The monitoring of HF patients remotely, or "telemedicine", has already been shown to influence outcomes amongst individuals with HF by improving quality of life, reducing HF-related hospitalizations, and optimizing the use of the limited resources in this field [18–22]. For example, a wireless implantable hemodynamic monitoring system of PAP allowed better remote HF management and reduced hospitalization rates [23, 24]. The current study extends these findings by showing that voice signal analysis used to identify and quantify a pre-

specified voice biomarker could form an additional method of remotely monitoring patients with HF. Further, by showing a very good agreement in the voice biomarker between separate voice recordings in each individual participant, we show that this method is capable of providing stable and reliable measurements. Thus, voice analysis, together with other advances in telecommunication technologies, could be used as adjuncts in the medical management of patients with HF, and could potentially have a significant impact in resource poor settings or in those with overburdened healthcare systems.

## Study limitations

This study has a number of limitations. First, this study reports association and does not provide evidence on a potential underlying mechanism. Second, this is a preliminary observational study that included a relatively homogenous population and thus further studies in larger and more diverse populations are required to ensure the generalizability of our findings. Third, all voice recordings were performed in the English language. There is a need for future studies to validate the consistency of our findings in other languages. Fourth, we did not assess for temporal changes in voice signal after implementing therapy. Last, some of our analyses were limited by sample size, particularly after stratification, and thus further larger studies will be required going forward.

## Conclusion

The current study shows an association between a non-invasive vocal biomarker derived from voice signal analysis and invasively derived hemodynamic indices related to PH obtained during clinically indicated cardiac catheterization. These results may have important and practical clinical implications for telemedicine and remote monitoring of patients with HF and PH.

## Supporting information

**S1 Dataset.**
(XLSX)

## Author Contributions

**Data curation:** Jaskanwal Deep Singh Sara, Elad Maor.

**Formal analysis:** Jaskanwal Deep Singh Sara, Elad Maor, Bradley R. Lewis.

**Funding acquisition:** Amir Lerman.

**Investigation:** Jaskanwal Deep Singh Sara, Elad Maor, Barry Borlaug, Diana Orbelo, Lliach O. Lerman, Amir Lerman.

**Supervision:** Amir Lerman.

**Writing – original draft:** Jaskanwal Deep Singh Sara.

**Writing – review & editing:** Jaskanwal Deep Singh Sara, Elad Maor, Barry Borlaug, Bradley R. Lewis, Diana Orbelo, Lliach O. Lerman, Amir Lerman.

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
