## [Decision Letter · Decision Letter 0]

29 Nov 2019

PONE-D-19-27216

Non-Invasive Vocal Biomarker is Associated with Pulmonary Hypertension

PLOS ONE

Dear Dr. Lerman,

Thank you for submitting your manuscript to PLOS ONE. After careful consideration, we feel that it has merit but does not fully meet PLOS ONE’s publication criteria as it currently stands. Therefore, we invite you to submit a revised version of the manuscript that addresses the points raised during the review process.

ACADEMIC EDITOR: All issues raised by reviewers are required.

We would appreciate receiving your revised manuscript by Jan 03 2020 11:59PM. To enhance the reproducibility of your results, we recommend that if applicable you deposit your laboratory protocols in protocols.io, where a protocol can be assigned its own identifier (DOI) such that it can be cited independently in the future. For instructions see: http://journals.plos.org/plosone/s/submission-guidelines#loc-laboratory-protocols

We look forward to receiving your revised manuscript.

Kind regards,

Vincenzo Lionetti, M.D., PhD

Academic Editor

PLOS ONE

**Journal Requirements:**

Dr. Elad Maor serves as a consultant for Beyond Verbal Communications.  The remaining authors have nothing to disclose

We note that one or more of the authors are employed by a commercial company: Beyond Verbal Communications

Reviewers' comments:

Reviewer's Responses to Questions

**Comments to the Author**

1. Is the manuscript technically sound, and do the data support the conclusions?

Reviewer #1: Partly

2. Has the statistical analysis been performed appropriately and rigorously? 

Reviewer #1: No

3. Have the authors made all data underlying the findings in their manuscript fully available?

Reviewer #1: No

4. Is the manuscript presented in an intelligible fashion and written in standard English?

Reviewer #1: Yes

5. Review Comments to the Author

Reviewer #1: The authors propose to use a voice biomarker to examine an association between levels of the voice biomarker and HF.

The voice biomarker is an unbounded scalar. The authors mention using a univariate analysis and also a multivariate analysis (both using repeated measures) on the data. However, it is unclear which analytic method is used. Is it some form of linear regression (since the outcome voice biomarker is a scalar)? Is it some form of logistic regression (since the authors report odds ratios)? Is it some form of generalized estimating equations (since the authors mention using repeated measures)? It is unclear which method was used for the univariate & multivariate analyses. The authors need to state the method used before a reviewer can determine if it is appropriate or not.

Additionally, the authors state that they used Student's t-test to analyze normally distributed, continuous data. However, from the boxplot for < 1.7 in Figure 2B, it appears that data is highly skewed and may not be normally distributed. Did the authors check the normality assumptions before running the student's t-tests? If so, that should be stated. If not, it should be done, and any transformations needed should be performed. If after transforming, the data is still not normally distributed, then a Mann-Whitney test should be used instead.

For all figures/tables, the meaning of * should be stated in the table/figure description. Additionally, for tables, it should be stated what values are present in the 2nd column (mean plus/minus SD, or percent of total).

6. PLOS authors have the option to publish the peer review history of their article (what does this mean?). If published, this will include your full peer review and any attached files.

Reviewer #1: No

---

## [Author Response · Author response to Decision Letter 0]

23 Dec 2019

Vincenzo Lionetti, M.D., PhD

Academic Editor

PLOS ONE

Manuscript. Ref. No.: PONE-D-19-27216 

Title: Non-Invasive Vocal Biomarker is Associated with Pulmonary Hypertension

Dear Dr. Lionetti,

Thank you very much for the thoughtful review of our manuscript. We hope that we are able to adequately address the reviewer’s comments and that you will consider the paper acceptable for publication.

Please find our responses to the reviewer’s comments below:

Journal Requirements:

We thank the editor for the comment. We have revised the manuscript accordingly.

Thank you for stating the following in the Financial Disclosure section:

Dr. Elad Maor serves as a consultant for Beyond Verbal Communications. The remaining authors have nothing to disclose

We note that one or more of the authors are employed by a commercial company: Beyond Verbal Communications

We thank the editor for the comment. Please note that Dr. Elad Maor serves as a consultant for Beyond Verbal Communications, but does not receive any salary from the company. Beyond Verbal Communications provided some of the funding for the study, but did not play a role in the study design, data collection and analysis, decision to publish, or preparation of the manuscript, and only provided financial support in the form of study funding. As such we have modified our Funding Statement as follows:

“This study was in part funded by Beyond Verbal Communications. The funders had no role in study design, data collection and analysis, decision to publish, or preparation of the manuscript. Dr. Elad Maor serves as a consultant for Beyond Verbal Communications. This author received no specific funding for this work. The remaining authors have nothing to disclose.”

We thank the editor for the comment. Please note that Dr. Elad Maor serves as a consultant for Beyond Verbal Communications, but does not receive any salary from the company. Beyond Verbal Communications provided some of the funding for the study, but did not play a role in the study design, data collection and analysis, decision to publish, or preparation of the manuscript, and only provided financial support in the form of study funding. As such we have modified our Competing Interests Statement as follows:

“Dr. Elad Maor serves as a consultant for Beyond Verbal Communications. This author receives no salary for this work. The remaining authors have nothing to disclose. This study was in part funded by Beyond Verbal Communications, who had no role in study design, data collection and analysis, decision to publish, or preparation of the manuscript.”

We thank the editor for this comment. Please note we have submitted our minimal dataset as a Supporting Information file, entitled S1 Dataset.

Reviewers' comments:

Reviewer's Responses to Questions

Comments to the Author

1. Is the manuscript technically sound, and do the data support the conclusions?

Reviewer #1: Partly

2. Has the statistical analysis been performed appropriately and rigorously? 

Reviewer #1: No

3. Have the authors made all data underlying the findings in their manuscript fully available?

Reviewer #1: No

4. Is the manuscript presented in an intelligible fashion and written in standard English?

Reviewer #1: Yes

5. Review Comments to the Author

Reviewer #1: The authors propose to use a voice biomarker to examine an association between levels of the voice biomarker and HF.

The voice biomarker is an unbounded scalar. The authors mention using a univariate analysis and also a multivariate analysis (both using repeated measures) on the data. However, it is unclear which analytic method is used. Is it some form of linear regression (since the outcome voice biomarker is a scalar)? Is it some form of logistic regression (since the authors report odds ratios)? Is it some form of generalized estimating equations (since the authors mention using repeated measures)? It is unclear which method was used for the univariate & multivariate analyses. The authors need to state the method used before a reviewer can determine if it is appropriate or not.

We thank the reviewer for the comment. The primary outcome of interest in the current study was the dependent variable moderate or severely elevated pulmonary arterial pressure, defined as a pulmonary artery pressure ≥ 35 mmHg. As a secondary outcome, we also looked at elevated pulmonary vascular resistance, defined as ≥ 1.7 Wood Units. As such, the dependent variables in the current study were binary categorical variables, and so we used logistic regression to determine the odds of having a moderate or severely elevated pulmonary arterial pressure and the odds of having an elevated pulmonary vascular resistance with the vocal biomarker as the independent variable. We have included the following comments in our statistical analysis section of the materials and methods.

“Univariate logistic regression analyses were undertaken to determine the association between the voice biomarker, as the independent variable, and each of the following individually as categorical dependent variables: a high PAP, PVR, PWCP, RA pressure, and CI. Each association was examined in all patients and after stratifying by a high versus lower PCWP. The distinction between high and lower PCWP was chosen to distinguish PH that was “post-capillary” in etiology and therefore related to coexisting left HF versus that which was “pre-capillary” in etiology and therefore related to a primary vascular and/or lung pathology. Finally, multivariate logistic regression analyses with repeated measures were undertaken to determine the relationships between the voice biomarker, as the independent variable, and each of the following individually as categorical dependent variables: a high PAP, PVR, PWCP, RA pressure, and CI. Each association was examined in all patients and after stratifying by a high versus lower PCWP. Each analysis was adjusted for age, sex, hypertension, diabetes mellitus, and NYHA class as these factors are known to be associated with PH and/or HF and could therefore act as potential confounders.” 

Additionally, the authors state that they used Student's t-test to analyze normally distributed, continuous data. However, from the boxplot for < 1.7 in Figure 2B, it appears that data is highly skewed and may not be normally distributed. Did the authors check the normality assumptions before running the student's t-tests? If so, that should be stated. If not, it should be done, and any transformations needed should be performed. If after transforming, the data is still not normally distributed, then a Mann-Whitney test should be used instead.

We thank the reviewer for the comment. We have added the following comment to the statistical analysis sub-section of the materials and methods:

“Normal distribution and equal variance were checked by the Shapro-Wills test, and Levene’s test respectively for each variable.” 

For all figures/tables, the meaning of * should be stated in the table/figure description. Additionally, for tables, it should be stated what values are present in the 2nd column (mean plus/minus SD, or percent of total).

We thank the reviewer for the comments. We have amended the figure captions and tables accordingly.

 6. PLOS authors have the option to publish the peer review history of their article (what does this mean?). If published, this will include your full peer review and any attached files.

Do you want your identity to be public for this peer review? For information about this choice, including consent withdrawal, please see our Privacy Policy.

Reviewer #1: No

---

## [Decision Letter · Decision Letter 1]

14 Jan 2020

PONE-D-19-27216R1

Non-Invasive Vocal Biomarker is Associated with Pulmonary Hypertension

PLOS ONE

Dear Dr. Lerman,

Thank you for submitting your manuscript to PLOS ONE. After careful consideration, we feel that it has merit but does not fully meet PLOS ONE’s publication criteria as it currently stands. Therefore, we invite you to submit a revised version of the manuscript that addresses the points raised during the review process.

ACADEMIC EDITOR: In light of data provided in the revised version of the manuscript, some concerns regarding statistical analysis came out. Statistical issue should be addressed carefully. 

We would appreciate receiving your revised manuscript by Feb 28 2020 11:59PM. To enhance the reproducibility of your results, we recommend that if applicable you deposit your laboratory protocols in protocols.io, where a protocol can be assigned its own identifier (DOI) such that it can be cited independently in the future. For instructions see: http://journals.plos.org/plosone/s/submission-guidelines#loc-laboratory-protocols

We look forward to receiving your revised manuscript.

Kind regards,

Vincenzo Lionetti, M.D., PhD

Academic Editor

PLOS ONE

Reviewers' comments:

Reviewer's Responses to Questions

**Comments to the Author**

1. If the authors have adequately addressed your comments raised in a previous round of review and you feel that this manuscript is now acceptable for publication, you may indicate that here to bypass the “Comments to the Author” section, enter your conflict of interest statement in the “Confidential to Editor” section, and submit your "Accept" recommendation.

Reviewer #1: (No Response)

2. Is the manuscript technically sound, and do the data support the conclusions?

Reviewer #1: No

3. Has the statistical analysis been performed appropriately and rigorously? 

Reviewer #1: No

4. Have the authors made all data underlying the findings in their manuscript fully available?

Reviewer #1: Yes

5. Is the manuscript presented in an intelligible fashion and written in standard English?

Reviewer #1: Yes

6. Review Comments to the Author

Reviewer #1: This reviewer thanks the authors for submitting the data as supplemental information as well as the clarification of methods used. Based on the data provided, it is not clear that the analysis performed is appropriate. The data provided contain information from at most three separate recordings. The authors do not mention how they deal with this repeated measures data in the analysis. This needs to be corrected and dealt with. It is highly suggested that the authors consult with a statistician before proceeding.

Student's t-tests are used. However, this is not the appropriate method for repeated measures data. Paired t-tests (when only 2 time points are present) or a 2-way repeated measures ANOVA are appropriate. IF the authors averaged the data at the three (or sometimes 2) time points, this should instead be stated, and justified as to why that is allowable/appropriate here. The text should also be modified if the values are averaged, as the voice characteristics section makes a point that each of the three recordings are distinct for a reason. If only one of the voice recordings was used for the analysis, the exact recording used should be stated instead.

It is noted that some patients do not have high quality recordings for all three time points. They instead only have 2 measurements. No mention is given to how this is handled in the analysis (are only 2 values averaged or is that patient removed from the analysis?). Additionally, some patients do not have a PVR value. No mention is provided as to how this is handled. Is the dataset for the primary outcome different from the dataset for the secondary outcome? If so, this should be stated, and mentioned how they differ. The alternate is to use the same (reduced) dataset for both the primary outcome and secondary outcome.

There is no such method as a multivariate logistic regression with repeated measures. The use of this term should be corrected to the appropriate method actually used. Logistic regression does not account for repeated measures, and it is not possible for that specific method to take them into account. Other methods can. If those were used, that needs to be indicated.

From looking at the results, it appears that the (up to) 3 biomarker values were averaged, and a single model for the primary outcome and different model for the secondary outcome was created. If this is correct, this needs to be stated and HIGHLY justified as it is not frequently appropriate to average across different recordings when each means something different.

A mixed effects model or generalized estimating equation approach might be more appropriate here since there are repeated measurements. Alternately, the authors might consider doing an analysis separately on each time recording (i.e. an analysis for recording 1, an analysis for recording 2, and an analysis for recording 3). This is where consulting with a statistician will be beneficial as they can best guide based on the specific questions and data at hand.

7. PLOS authors have the option to publish the peer review history of their article (what does this mean?). If published, this will include your full peer review and any attached files.

Reviewer #1: No

---

## [Author Response · Author response to Decision Letter 1]

24 Feb 2020

Comments to the Author

1. If the authors have adequately addressed your comments raised in a previous round of review and you feel that this manuscript is now acceptable for publication, you may indicate that here to bypass the “Comments to the Author” section, enter your conflict of interest statement in the “Confidential to Editor” section, and submit your "Accept" recommendation.

Reviewer #1: (No Response)

2. Is the manuscript technically sound, and do the data support the conclusions?

Reviewer #1: No

3. Has the statistical analysis been performed appropriately and rigorously? 

Reviewer #1: No

4. Have the authors made all data underlying the findings in their manuscript fully available?

Reviewer #1: Yes

5. Is the manuscript presented in an intelligible fashion and written in standard English?

Reviewer #1: Yes

6. Review Comments to the Author

Reviewer #1: This reviewer thanks the authors for submitting the data as supplemental information as well as the clarification of methods used. Based on the data provided, it is not clear that the analysis performed is appropriate. The data provided contain information from at most three separate recordings. The authors do not mention how they deal with this repeated measures data in the analysis. This needs to be corrected and dealt with. It is highly suggested that the authors consult with a statistician before proceeding.

We thank the reviewer for the comment. We have consulted with a statistician, who has been added to the manuscript as a co-author. We agree that our previous statistical methods may have erroneously inflated the power of our analyses, as we had undertaken the primary analyses for all separate individual voice recordings. We have changed the analysis now by calculating the mean voice biomarker value from the three separate recordings for each individual study participant, and have used this mean value in the subsequent analyses. We elected the purse the mean for two reasons: we found very good agreement between repeated recordings amongst our cohort (the intra-class correlation coefficient was 0.829 (95% CI 0.740 – 0.889) showing that our repeated voice biomarker values were stable and reliable and in turn correlated well with each other. Thus the mean provides a reasonable summary statistic for the multiple recordings. Secondly, in our previous study (in press) in which we showed an association between the same voice biomarker and HF outcomes including rehosptalization, we did not show any differences in the association between the voice biomarker and clinical outcomes when the voice biomarker used was derived from individuals recording their voices talking about positive, negative, or neutral experiences separately. Thus there did not seem to be a reasonable necessity on which to test the association between the voice biomarker derived under the three separate circumstances and the indices related to pulmonary hypertension in the current study. Nevertheless, we do present data in the results section of the manuscript showing the association between the voice biomarker derived whilst talking about a negative, positive and neutral experience and mean pulmonary artery pressure and pulmonary vascular resistance. We were able to do this in a univariable analyses but not multivariable analyses due to limitations of sample sizes. Please see below for the additions to the manuscript:

After excluding poor quality recordings, values of the pre-specified voice biomarker were obtained from each high quality recording for each patient. In our recent study in which we showed that the same pre-specified voice biomarker was associated with increased mortality and re-hospitalization in patients with HF [28] (manuscript in press), we did not show any significant differences in the association between the voice biomarker and clinical outcomes when the voice biomarker used was derived from individuals recording their voices talking about positive, negative, or neutral experiences separately. Consequently, we elected to determine the agreement across the separate voice biomarker values for each individual participant by calculating the intra-class correlation coefficient with 95% confidence interval. We then calculated the mean voice biomarker value for each patient and used these numbers in our final analyses. The mean biomarker values were then compared between groups using Student’s t-test. 

A pre-established vocal biomarker was determined based on each separate recording for each individual participant. The intra-class correlation co-efficient between the separate voice recording biomarker values for each individual participant was 0.829 (95% CI 0.740 – 0.889) implying very good agreement between values. Thus, the mean biomarker was calculated for each patient. 

Table 2 shows the results of a univariable analysis evaluating the association between hemodynamic indices and the mean voice biomarker among all patients and after stratifying by high versus low PCWP. In all patients, the mean voice biomarker was significantly associated with a mean PAP ≥ 35 mmHg such that an increase in the mean biomarker by 1 unit was associated with an odds ratio (95% CI) of 1.92 (1.00 – 3.65, p=0.049). This association did not persist after stratifying by high or low PCWP. In a separate univariable analysis, the biomarker derived while describing a negative experience was significantly associated with a high mean PAP, odds ratio (95% CI) 1.91 (1.05 – 3.49, p=0.035) but not when the biomarker was derived while describing a neutral experience, odds ratio (95% CI) 1.31 (0.69 – 2.49, p=414) or positive experience, odds ratio (95% CI) 1.53 (0.85 – 2.76, p=016). The mean voice biomarker was not significantly associated with a PVR ≥ 1.7 Wood Units amongst all patients or in those after stratifying by high and low PCWP. Further, the voice biomarker was not significantly associated with a high PVR when it was derived in individuals describing a negative experience, odds ratio (95% CI) 1.54 (0.88 – 2.71, p=0.13), neutral experience, odds ratio (95% CI) 1.57 (0.76 – 3.23, p=0.223) or positive experience, odds ratio (95% CI) 1.44 (0.75 – 2.75, p=0.272). The voice biomarker was not significantly associated with any other hemodynamic index. 

Student's t-tests are used. However, this is not the appropriate method for repeated measures data. Paired t-tests (when only 2 time points are present) or a 2-way repeated measures ANOVA are appropriate. IF the authors averaged the data at the three (or sometimes 2) time points, this should instead be stated, and justified as to why that is allowable/appropriate here. The text should also be modified if the values are averaged, as the voice characteristics section makes a point that each of the three recordings are distinct for a reason. If only one of the voice recordings was used for the analysis, the exact recording used should be stated instead.

We thank the reviewer for the comment. We elected to calculate the mean value of the voice biomarker for each individual participant in the cohort. Please see above for details.

It is noted that some patients do not have high quality recordings for all three time points. They instead only have 2 measurements. No mention is given to how this is handled in the analysis (are only 2 values averaged or is that patient removed from the analysis?). Additionally, some patients do not have a PVR value. No mention is provided as to how this is handled. Is the dataset for the primary outcome different from the dataset for the secondary outcome? If so, this should be stated, and mentioned how they differ. The alternate is to use the same (reduced) dataset for both the primary outcome and secondary outcome.

We thank the reviewer for the comment. We have included the following statement to clarify our statistical methods:

In cases in which a study participant had one or more voice recording samples excluded due to poor quality, the remaining high-quality samples were retained and used to determine the mean biomarker value for that patient. 

The same dataset was used for all analyses. 

There is no such method as a multivariate logistic regression with repeated measures. The use of this term should be corrected to the appropriate method actually used. Logistic regression does not account for repeated measures, and it is not possible for that specific method to take them into account. Other methods can. If those were used, that needs to be indicated.

We thank the reviewer for the comment. We agree and thus have removed this statement from the manuscript. We have elected to undertake our analyses using the mean value of the voice biomarker for each patient, and then undertook univariable and multivariable logistic regression analyses using these values. See above for more details. 

From looking at the results, it appears that the (up to) 3 biomarker values were averaged, and a single model for the primary outcome and different model for the secondary outcome was created. If this is correct, this needs to be stated and HIGHLY justified as it is not frequently appropriate to average across different recordings when each means something different.

We thank the reviewer for the comment. Please see above for more details. We elected to determine the mean value of the voice biomarker for each individual participant, because the mean value provided a reasonable and appropriate summary statistic for each patient given that the intra-class correlation coefficient that we calculated showed good agreement between values. This provided evidence that the separate voice recordings for each patient were stable and reliable, and therefore suitable to combine to obtain a mean value. Furthermore our previous study in this area did not show that voice biomarker values obtained under different circumstances (describing a neutral, positive or negative situation) were differently associated with heart failure clinical outcomes, and thus we did not intend to primarily test the different associations between the voice biomarker under different circumstances and the invasive indices measured in this study. Thus, we elected to use the mean values in our subsequent primary analyses, and consequently have changed our final results and figures accordingly. In addition, we assessed the association between the mean voice biomarker value and mean PAP and PVR separately in two different models. Mean PAP was our primary outcome of interest as it is the marker that facilitates the diagnosis of pulmonary hypertension itself and characterizes its severity. PVR is a useful adjunct measure when looking at pulmonary hypertension, but is not of primary importance when making the diagnosis or determining the severity of pulmonary hypertension. Further, these two measures are related to each other and we did not want to have them present in the same model as both markers could account for at least some of the same proportion of variability of the voice biomarker across different values.

A mixed effects model or generalized estimating equation approach might be more appropriate here since there are repeated measurements. Alternately, the authors might consider doing an analysis separately on each time recording (i.e. an analysis for recording 1, an analysis for recording 2, and an analysis for recording 3). This is where consulting with a statistician will be beneficial as they can best guide based on the specific questions and data at hand.

We thank the reviewer for the comment. Please see above for more details. We describe the results of the association between the mean value of the voice biomarker and mean PAP and PVR. We also show in our results section the findings of the association between the value of the voice marker derived when individuals are describing a negative experience, positive experience, and neutral experience and mean PAP and PVR in univariable analyse..

---

## [Decision Letter · Decision Letter 2]

25 Mar 2020

Non-Invasive Vocal Biomarker is Associated with Pulmonary Hypertension

PONE-D-19-27216R2

Dear Dr. Lerman,

We are pleased to inform you that your manuscript has been judged scientifically suitable for publication and will be formally accepted for publication once it complies with all outstanding technical requirements.

With kind regards,

Vincenzo Lionetti, M.D., PhD

Academic Editor

PLOS ONE

Additional Editor Comments (optional):

Reviewers' comments:

Reviewer's Responses to Questions

**Comments to the Author**

1. If the authors have adequately addressed your comments raised in a previous round of review and you feel that this manuscript is now acceptable for publication, you may indicate that here to bypass the “Comments to the Author” section, enter your conflict of interest statement in the “Confidential to Editor” section, and submit your "Accept" recommendation.

Reviewer #1: All comments have been addressed

2. Is the manuscript technically sound, and do the data support the conclusions?

Reviewer #1: Yes

3. Has the statistical analysis been performed appropriately and rigorously? 

Reviewer #1: Yes

4. Have the authors made all data underlying the findings in their manuscript fully available?

Reviewer #1: Yes

5. Is the manuscript presented in an intelligible fashion and written in standard English?

Reviewer #1: Yes

6. Review Comments to the Author

Reviewer #1: (No Response)

7. PLOS authors have the option to publish the peer review history of their article (what does this mean?). If published, this will include your full peer review and any attached files.

Reviewer #1: No

---

## [Editor Report · Acceptance letter]

2 Apr 2020

PONE-D-19-27216R2 

Non-Invasive Vocal Biomarker is Associated with Pulmonary Hypertension 

Dear Dr. Lerman:

I am pleased to inform you that your manuscript has been deemed suitable for publication in PLOS ONE. Congratulations! Your manuscript is now with our production department. 

With kind regards,

on behalf of

Prof. Vincenzo Lionetti 

Academic Editor

PLOS ONE